# Parametrization Study for Optimal Pre-Combustion Integration of Membrane Processes in BIGCC

Maytham Alabid [1] and Cristian Dinca [1,2,*]

1    Faculty of Energy, University Politehnica of Bucharest, Splaiul Independenței, 060042 Bucharest, Romania
2    Academy of Romanian Scientists, Ilfov 3, 050044 Bucharest, Romania
*    Correspondence: crisflor75@yahoo.com

**Abstract:** Presently, the utilization of biomass as an energy source has gained significant attention globally due to its capacity to provide constant feedstock. In 2020, biomass combustion generated 19 Mt of $CO_2$, representing an increase of 16% from the previous year. The increase in $CO_2$ emissions is fundamentally due to biomass gasification in power plants. Due to the growing demand to reduce greenhouse gas emissions, this paper aims to improve $CO_2$ capture technologies to face this challenge. In this context, the utilization of three stages of the polymer membrane process, using different compressor pressure values, has been technically and economically analyzed. The proposed solution was combined pre-combustion in a BIGCC process equipped with a Siemens gas turbine with an installed power capacity of 50 MW. The article simulated energy operations by using membranes of polymer and CHEMCAD software improved in the $CO_2$ integration research project. Consequently, polymeric membranes with $CO_2$ permeability of 1000 GPU were examined while $CO_2$ selectivity towards nitrogen was investigated to be 50. It was observed that by increasing the surface area of the polymer membrane (400,000–1,200,000 $m^2$) an increase of 37% occurs in $CO_2$ capture efficiency. On the other hand, LCOE increased from 97 to 141 EUR/MWh. The avoided cost of $CO_2$ captured was 52.9 EUR/ton.

**Keywords:** pre-combustion; membrane; $CO_2$ capture system; gasification; process integration

## 1. Introduction

Living standards and economic booms are usually evaluated by per capita energy utilization [1]. As a result of the increase in population and the development of living conditions, energy requirements (especially electricity energy) are continuously increasing. Currently, most of the electricity generated around the world is produced from large-scale fossil fuels, particularly coal [2]. The main source of climate change is the high $CO_2$ emissions from these fossil fuel power plants. In recent years, intensive studies have been conducted to handle this challenge to human existence on the planet [3,4]. Carbon capture and storage (CCS) is the current potential preference for using fossil fuels for wide-ranging power generation with no climate impact [5,6]. To reduce the climate change impact, several technological methods for $CO_2$ recovery are accessible at various maturity scales [7]. Presently, different $CO_2$ recovery processes have been enhanced, such as post-combustion [8], pre-combustion [9], or oxy-fuel combustion which generates a large amount of $CO_2$ stream [10].

After being captured, $CO_2$ is transported to preserve it safely underground or to be used in many different industrial processes [11]. The essential drawbacks of using CCS technologies in power plants are the energy consumption demands and the penalization of the plant's overall efficiency [12–14].

The chemical absorption process is the classical process to capture $CO_2$ that requires high thermal energy, high capital, and operating costs. Degradation, solvent emissions, and other environmental troubles are also some of the disadvantages of this process. As a

result, another more efficient and promising technology can be used to capture $CO_2$, such as membranes. Generally, a membrane is a thin film used to separate two phases [15–17]. Until now, several membranes have been promoted that are recognized for their high permeability and selectivity for better achievement in $CO_2$ capture. Various types of membrane materials are utilized for the carbon separation process, such as common polymers, sieve membranes, and inorganic membranes [18,19]. The high electric energy required for the membrane $CO_2$ capture process, transfer and storage operations, and the large size demanded in reserving $CO_2$ are the prime drawbacks of incorporating membrane technology into power plants. Indeed, to contend with the chemical absorption technology regarding the cost, a specific membrane combination system is required to reduce the membrane surface and electricity needed. Various articles have been published in terms of the membrane $CO_2$ capture process with main variables that influence the recovery execution of carbon dioxide and its pureness [20–22].

Yang et al. (2009) examined the $CO_2$ removal process from a flue gas flow of 11.57 m³ (STP)/s with a 15% carbon dioxide concentration. Different $CO_2$ permeabilities have been examined to achieve 90% $CO_2$ recovery and 95% purity. The author revealed that the pressure ratio across the membrane stage has a significant impact on $CO_2$ removal performance, where the optimal result had been obtained at a particular pressure difference. In addition, the author found that membrane process can defeat the current gas separation technology (chemical absorption) concerning the investment cost.

Brunetti et al. (2010) presented a simple carbon dioxide capture process with a 13% $CO_2$ composition. The author found that the membrane characteristics and other parameters directly impact the process achievement of recovery rate and purity of $CO_2$. For a specific gas flow, membrane size, and pressure difference across the membrane, the author declared that the lower the capture process, the higher the purity of the captured $CO_2$, and vice versa. More membrane stages were recommended to increase $CO_2$ removal concentration due to low carbon dioxide composition (15%), even with high-pressure utilization. For multiple stages of membrane, doubling the pressure ratio improved the removal process of carbon dioxide by two to three times and obtained better results regarding the purity of $CO_2$ removed.

Zhang et al. (2013) assessed a membrane technology integrated into a coal-fired power plant. The author investigated the effect of membrane performance and configuration on membrane area, power required, and recovery price. The results revealed that an increase in the total $CO_2$ recovered was achieved by enlarging membrane surface, while the concentration of carbon dioxide removed was increased with a significant reduction in the membrane surface. Thus, the author managed to distribute the flue gas between at least two stages of membrane which helps to increase the capture execution. For an optimal point of $CO_2$ recovery process, several parameters of a membrane configuration must be analyzed to select the optimum.

The different studies presented above demonstrate the main parameters that should be varied for a specific membrane's different stages configuration to obtain the optimal case regarding $CO_2$ recovery performance and carbon dioxide permeate concentration.

Alternative solutions to meet the energy requirements with poor/no $CO_2$ release are either by raising energy efficiency to decrease the $CO_2$ content in the flue gas or by using a $CO_2$-neutral fuel, like biomass [7,23].

The fundamental feature of using biomass is the fact that biomass absorbs $CO_2$ during its growth, which is equal to that produced in the combustion step. The most efficient path to utilize biomass is by gasification [24,25]. Integration of a biomass gasification method together with an efficient combined cycle power plant is a promising potential choice for $CO_2$-neutral energy production [26]. The energy losses for the $CO_2$ recovery process are recompensed by extra carbon dioxide capture, producing a fine $CO_2$-negative power plant. However, the gasification method of using biomass as fuel and the combined cycle with the carbon removal operation, is a potentially promising technology to meet the carbon reduction goal to face the threats of climate change [26,27].

In the state of solid fuels (as biomass), the pre-combustion recovery method is preferable due to both the carbon dioxide molecules in the syngas (more than 20%) and the pressure of the gas (20–50 bar). These values could be obtained using $O_2$ instead of air, which is better than the state of post-combustion recovery [28]. A classical pre-combustion $CO_2$ removal system needs a gasification section, as shown in Figure 1. In terms of the gasification process, solid fuel is modified to syngas enriched with CO and $H_2$. After particulate elimination through a cyclone separation section, syngas is then sent to the water gas shift (WGS) section, where carbon monoxide interacts with the vapor to produce a mixture of $CO_2$ and $H_2$ [29]. Then, the mixture is processed in desulphurization and carbon dioxide separation methods (e.g., membrane), generating a fuel full of $H_2$ that can be utilized in several ways, for instance in gas turbines or interior burning engines [30,31].

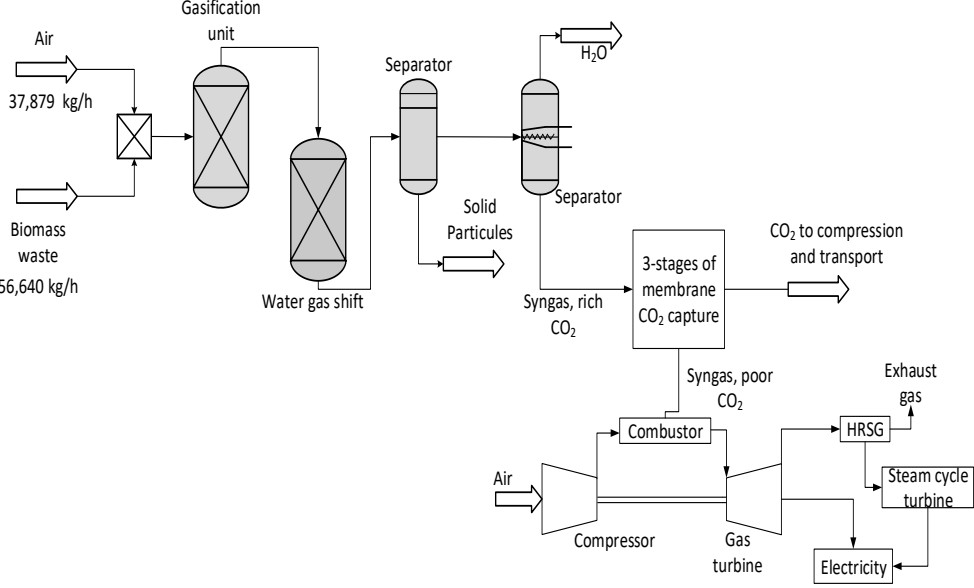

**Figure 1.** Scheme diagram of BIGCC with pre-combustion carbon capture.

The current paper is focused on the pre-combustion carbon dioxide capture technology using a membrane process applied to a power plant of 50 MW using biomass as a fuel. Different parameters (e.g., compressor pressure) values have been examined to obtain 90% of $CO_2$ capture efficiency and purity of no less than 95%.

## 2. Gasification of Biomass Power Plant

In this section, the characteristics and procedure of biomass gasification in a biomass power plant, that uses gas and steam turbine to generate 50 MW, with and without the membrane carbon capture system (CCS), are presented. Figure 1 below shows the scheme of BIGCC with a pre-combustion $CO_2$ recovery standard.

### 2.1. Gasification of Biomass without CCS

The biomass gasification method can be introduced by converting solid or liquid biofuel to syngas. Syngas has elevated thermal energy that can be used directly to produce both heat and electricity. Typically, the major elements of biomass gasification are the substance, gasification component, water–gas reactor, and separator unit to remove sulfur and other impurities, such as dust [32].

Partial oxidation is the process of converting biomass into synthesis gas. This operation occurs with the attendance of air or oxygen, a separation component needed to produce pure $O_2$ from air in the second situation, generating mainly CO and $H_2$ (syngas). In addition to the low production of $CH_4$, $CO_2$, $N_2$, and hydrocarbons (e.g., ethane) [33–36]. Unwanted gases, such as $H_2S$, can be produced as well, which can be eliminated by using a desulphurization unit—see Figure 1. Usually, the existence of such gases can be

generated based on the functional status of the gasification. The appropriate feedstock for partial oxidation must have a moisture content of less than 35%. A drying or preheating treatment is suggested for substances with high moisture before entering the gasifier, due to the damage that might be caused to the total process efficiency [37,38]. According to reference [39], the biomass material, the gasification process, and the operation status are the main factors that directly impact the low heating value (LHV), which varies from 4–13 MJ/Nm$^3$.

The biomass feedstock considered in this research was plum pits, the final and proximate investigations of the plum pits utilized in the current evaluation are presented in Table 1. A 56,640 kg/h waste flow rate was introduced with air as a carrier gasifier into the gasification unit at a temperature range of 500–1400 °C, see Figure 1. Air was assumed as a gasification agent in this current process to reduce the cost of using pure oxygen, this value selection was discussed further in detail. Then, a water–gas shift reactor was applied to the syngas flow where most of the CO was converted to carbon dioxide by the reaction with steam. Furthermore, the gas stream was discharged to a series of separator units to remove solid particles (e.g., ash) and moisture, and then the syngas was introduced into the carbon capture technology (membrane) to remove $CO_2$. All operations were planned and simulated by utilizing CHEMCAD software version (8.1).

**Table 1.** Plum pits main characteristics [36].

| Biomass Waste Component | Value (%) |
|---|---|
| Carbon | 49.21 |
| Oxygen | 41.81 |
| Hydrogen | 6.61 |
| Ash | 1.4 |
| Nitrogen | 0.89 |
| Sulfur | 0.08 |
| LHV | 18,939.1 kJ/kg |

The syngas composition produced from the biomass waste (plum pits) after the separators section regarding the steady-state situations and the final and proximate investigations is shown in Table 2, where the data are gained by the CHEMCAD simulation program (as mentioned previously). Different numbers (0.15–0.45) of equivalent ratios (ER) have been performed to obtain the peak value for Cold Gas Efficiency (CGE) from the separator. These calculations are carried out to define the amount of syngas flow rate entering the capture process. The optimum value of CGE was around 42%, which was obtained at an ER of 25%. The various ER assumptions, their results, and the equations used will be discussed further.

**Table 2.** Syngas composition after the separators section.

| Component | Unit | Value |
|---|---|---|
| $N_2$ | %mole | 43.66 |
| $H_2$ | %mole | 30.73 |
| $CO_2$ | %mole | 23.03 |
| CO | %mole | 2.58 |
| Syngas flow | kmol/h | 3109 (ER = 25%) |
| LHV | kJ/kg | 3441 |
| Temperature | °C | 40 |
| Pressure | bar | 1.013 |

### 2.2. Integration of Membrane $CO_2$ Capture Process

The Biomass gasification process was equipped with pre-combustion carbon capture by using membrane technology. The syngas generated from the gasification operation was integrated into a new scheme consisting of different membrane stages in order to separate $CO_2$ with the highest possible amount of $CO_2$ purity.

Polymeric membranes are well investigated for different applications to remove $CO_2$ from flue gas on account of their soft manufacturability and asymmetrical frames, which help the high flux of membrane to be ready for large-scale implementations [40]. Solution diffusion is a qualified mechanism which captures carbon dioxide via polymeric membrane where the achievements of the membrane can be improved through developing the diffusion and/or sorption qualities. Convection and diffusion are the main indicators that guide the $CO_2$ transport from flue gas to the membrane, where diffusion of carbon dioxide within the polymeric membrane as a result of the concentration gradient that has been created [41,42]. The carbon-dioxide-rich gas passes across the membrane pipe module, where a convection mechanism occurs as a result of mass transfer. Furthermore, the length of the pipe causes the diffusion mechanism [43]. Concerning the two methods above, $CO_2$ is captured and removed from the syngas stream.

Although one single stage would provide a sufficiently high capture efficiency, $CO_2$ purity remains quite low [7]. Thus, three stages of the membrane unit were considered to capture 90% of $CO_2$ emissions with a minimum of 95% carbon dioxide purity.

The membrane scheme simulated in this paper is shown in Figure 2 below. The scheme presents in detail the input and outputs of the main data with the ancillary components (e.g., compressors).

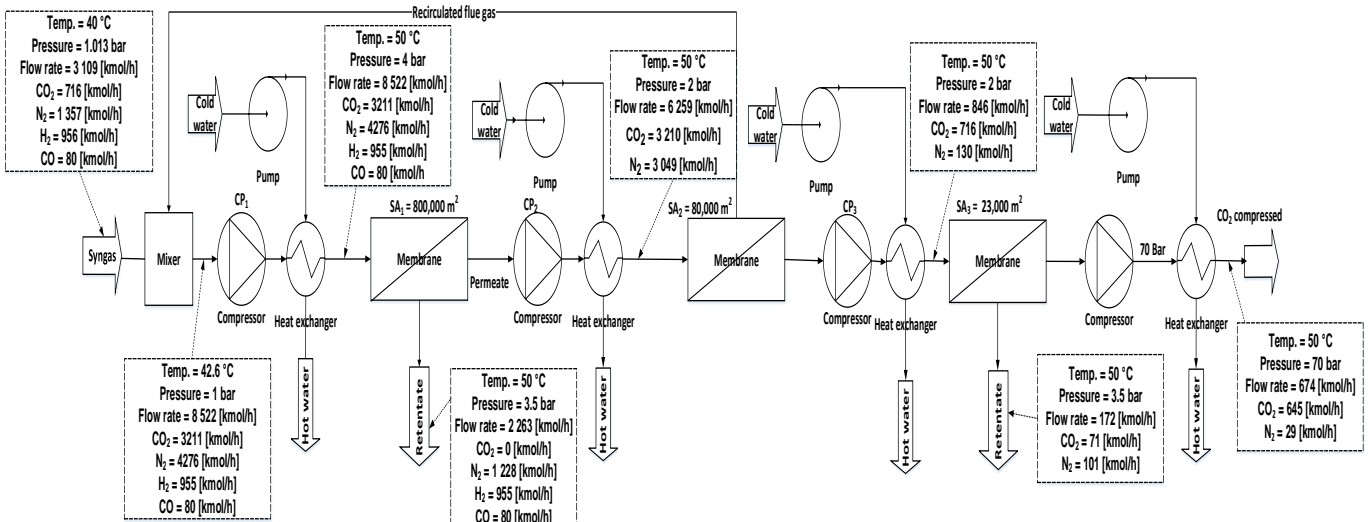

**Figure 2.** Three stages of the membrane scheme with different components.

The syngas feed characteristics before entering the membrane process are defined in Table 2. As presented in Figure 2, the other molecules of syngas (as $H_2$ and $N_2$) retained from the first and third membrane units were used for generating electricity via the gas and steam turbine. To increase the efficiency of the capture process, a recirculated line was assumed from the second membrane retentate side to decrease the syngas losses where it was combined with the primary syngas flow by a mixer. Prior to entering any membrane stage, the stream has to compress at a specific pressure to increase the efficiency of that membrane [44–46]. The high pressure applied to the stream leads to an increase in the temperature, which requires a cooling unit (heat exchanger) to reduce it, see Figure 2. The $CO_2$ captured flow, after the third membrane, was introduced to a compression unit with an elevated pressure (considered 70 bar) to provide compressed carbon dioxide that can be stored or used in many industries [47,48].

The current article proposes one configuration (three stages) to obtain a 90% capture rate and more than 95% carbon dioxide purity, where elevated purity is recommended for transport or other purposes, like methanol production [49]. As a rule, $CO_2$ capture efficiency basically depends on pressure difference as a driving force, membrane surface area, and $CO_2$ permeance, while purity relies on membrane selectivity, surface area, and

pressure values around the membrane. Since high purity demands a low membrane surface area, three stages have been proposed to increase both the efficiency and purity of carbon dioxide and to reduce the high electrical energy required for the recovery process.

As a result of the poor membrane material working period after 5 years, a replacement process has to be applied due to the low achievement [50–52]. An advanced process has been utilized in 2020 to integrate the CA enzyme into a polymer material called polyacrylamide (PSF 50 K) [53].

The prime target of this paper is to estimate the performance of membrane three-stage $CO_2$ capture integrated into a waste biomass power plant (plum pits) by the pre-combustion process. Different parameters have been used, such as compressor pressure and membrane surface area, see Table 3, to obtain the lowest electrical energy required for the main article goal (90% efficiency, no less than 95% purity of $CO_2$). For a membrane $CO_2$ capture process, low cost can be defined by decreasing ancillary power consumption [54]. The main sources of increased electric energy are syngas compressors and water pumps. However, for a particular carbon recovery efficiency, if the pressure around the first membrane unit is high, that leads to more electricity demand, more carbon dioxide purity, and less area for the membrane. On the other hand, when the pressure is low, a bigger surface area is required, less electricity is consumed, and a lower purity of the removed carbon is obtained.

**Table 3.** Indicators of membrane process used and power plant major parameters.

| Parameter | Unit | Value |
|---|---|---|
| Membrane type | - | Spiral wound |
| Flow pattern | - | Counter-current |
| $CO_2$ permeability | GPU | 1000 |
| $N_2$ permeability | GPU | 20 |
| $CO_2/N_2$ selectivity | - | 50 |
| Efficiency of compressors | % | 90 |
| Efficiency of pumps | % | 90 |
| Water pumps pressure | bar | 3 |
| Heat exchanger temperature out (All) | °C | 50 |
| First compressor pressure ($CP_1$) | bar | 2–6 |
| First membrane surface area ($MSA_1$) | m$^2$ | 400,000–1,200,000 |
| Second compressor pressure ($CP_2$) | bar | 2–6 |
| Second membrane surface area | m$^2$ | 80,000 |
| Third compressor pressure ($CP_3$) | bar | 2–6 |
| Third membrane surface area | m$^2$ | 23,000 |
| Power plant main parameters | | |
| The temperature of super-critical vapor | °C | 585 |
| The pressure of super-critical vapor | bar | 290 |
| LHV of the steam | kJ/kg | 17,139 |
| The net efficiency of the power plant (LHV biomass) | % | 29.8 |

Table 3 shows the essential characteristics and the variations of the membrane operation components, in addition to the essential parameters of the power plant. The permeability data in the table below were obtained from reference [13].

## 3. Technical and Economical Assessments

The article simulated different ER regarding the stoichiometric air in order to set the best CGE value, from which syngas flow amount can be chosen and defined in the membrane process, see Figure 1. Selecting the optimum value of CGE helps to define the mole fraction of plum pits content ($CO_2$, $N_2$, etc.).

Various $MSA_1$ and compression units are utilized to assess the technical and economic factors for the membrane $CO_2$ capture process used, where $MSA_1$ has the main effect on $CO_2$ capture rate and energy consumption. Compressors have a considerable influence on $CO_2$ purity, process efficiency, and power consumption. As mentioned, the CHEMCAD

program was utilized for all the examined values. Simulation for the membrane configuration (Figure 2) introduces all the substantial information, such as composition, mass flow rate, temperatures, pressures, syngas content, $CO_2$ purity, and electric power requirement. The next equations were harnessed to calculate the main plant-showing factors:

- Real air introduced in the gasifier shows the amount of equivalent ratio times the stoichiometric air introduced in the gasifier, and it is computed respecting the form [55]:

$$\text{Real air} = \text{ER} \times \text{stoich. air} \tag{1}$$

where the stoichiometric air value is 350,618 kg/h

- Cold gas efficiency (CGE) represents the total gasification operation efficiency, which can be calculated as follows [55]:

$$\text{CGE} = \frac{\text{syngas flow} \times \text{LHV}_{\text{syngas}}}{\text{Biomass flow} \times \text{LHV}_{\text{Biomass}}} \times 100\% \tag{2}$$

- Required power for the membrane process can be computed through the total electric energy consumed by the auxiliary membrane components.

$$\text{Membrane power consumption} = \sum \text{P}_{\text{ax}} \tag{3}$$

$\text{P}_{\text{ax}}$ is the amount of energy demanded, in kW, for the auxiliary units (e.g., compressors). The following parameters were utilized to count the economic estimation [11]:

- Levelized cost of electricity (LCOE), in EUR/kWh, can be determined by Equation (4) below:

$$\text{LCOE} = \frac{\text{CAPEX} + \text{OPEX}}{\text{W}_{\text{net}}} \tag{4}$$

where $\text{W}_{\text{net}}$ represents the net electric energy generation, in kWh.

- SPECCA, utilized for the membrane process, in MJ/kg, can be calculated regarding the next formula [56]:

$$\text{SPECCA} = \frac{3600 \times \left( \text{W}_{\text{net,NO capture}} - \text{W}_{\text{net, With capture}} \right)}{\left( \text{W}_{\text{net, No capture}} \times \text{E}_{\text{No capture}} \right) - \left( \text{W}_{\text{net, with capture}} \times \text{E}_{\text{with capture}} \right)} \tag{5}$$

where $\text{E}_{\text{No capture}}, \text{E}_{\text{with capture}}$ demonstrate the overall $CO_2$ emissions of the power plant with and without membrane $CO_2$ capture utilization, both in kg/kWh.

- Carbon dioxide capture cost ($CO_{2,CC}$) can be defined as the ratio of the plant electricity price difference with and without membrane usage per amount of $CO_2$ captured, in EUR/t, regarding the next formula:

$$CO_{2,\ CC} = \frac{\text{LCOE}_{\text{with capture}} - \text{LCOE}_{\text{No capture}}}{CO_2 \text{captured}} \tag{6}$$

- On the other hand, carbon dioxide avoided cost ($CO_{2,AC}$) is determined basically through the electricity price difference divided by $CO_2$ emissions variations with and without $CO_2$ capture use, in EUR/t, and the following formula presents that ratio:

$$CO_{2,\ AC} = \frac{\text{LCOE}_{\text{with capture}} - \text{LCOE}_{\text{No capture}}}{\text{E}_{\text{No capture}} - \text{E}_{\text{with capture}}} \tag{7}$$

In Table 4, the fundamental economic items used to calculate the different parameters, such as CAPEX, are defined below.

**Table 4.** The base factors regarding the economic assessment.

| Item | Unit | Value |
|------|------|-------|
| Project lifetime | years | 25 |
| Price of electric energy | EUR/MWh | 160 [57] |
| The price of gas turbine | MEUR | 93 [57] |
| The price of steam turbine | MEUR | 52 [58] |
| The price of Condenser | MEUR | 39 [58] |
| The price of HRSG | MEUR | 34 [58] |
| The price of Gasification unit | MEUR | 162 [58] |
| The reactor of water–gas shift | MEUR | 21.12 [58] |
| The price of separator | MEUR | 58 [58] |
| The price of ash treatment | MEUR | 16 [58] |
| $CO_2$ emissions fees | EUR /t | 66 [59] |
| Period of working | hour/year | 75% of 8760 |
| Indicator of Availability | % | 85 [58] |
| Rate of deduct | % | 8 [12] |
| Membrane process | | |
| Membrane unit particular price | EUR/m$^2$ | 50 [58] |
| The lifetime of membrane modules | years | 5 [52] |
| The price of pumps | EUR/kW | 1350 [58] |
| The price of compressors | EUR/kW | 1800 [58] |
| The price of a membrane alteration | EUR/m$^2$ | 10 [7] |
| Employments payment | EUR/hour | 15 [58] |
| Carbon dioxide compression stage | | |
| The price of $CO_2$ compressor unit | MEUR | 11.7 [58] |
| The price of cooling compressors | MEUR | 0.87 [58] |

For the present paper, the following indicators were determined to set as the project either profitable or the opposite.

- Net present value (NPV), in EUR, was computed regarding the formula:

$$NPV = \sum_{i=1}^{n_f} \frac{IN_i - C_i - A_i}{(1+r)^i} - \sum_{i=1}^{n_r} I_i \times (1+r)^i \qquad (8)$$

In which $IN_i$ demonstrates the actual bonus of the year i,

$C_i$ the amount of money required for maintenance for one year;
$A_i$ the value of a payback loan (if exists) for one year;
$I_i$ the actual investment for one year;
r the rate of deduction.

- Internal rate of return (IRR) was computed by respecting the next equation:

$$NPV = \sum_{i=1}^{n} \frac{IN_i - C_i - I_i}{(1+IRR)^i} = 0 \qquad (9)$$

since r = IRR for any investment venture, then NPV = 0.

- Equation (10) represents the formula to calculate the Discount payback period (DPP), in years:

$$NPV = \sum_{i=1}^{DPP} \frac{IN_i - C_i - I_i}{(1+r)^i} \qquad (10)$$

- For a decision on considering whether the project is financially well-planned, the profitability index (PI) is determined as the ratio of summation NPV and deduction of investment (IA) per deduct investment as follows:

$$PI = \frac{NPV + IA}{IA} \qquad (11)$$

## 4. Results and Discussion

Two different gasification processes were simulated in this article to produce syngas and three units of membranes to remove $CO_2$ from the syngas produced.

1. The gasification process

Several equivalent ratios have been simulated (0.15–0.45) to obtain the best cold gas efficiency after the separator units at 40 °C, the chosen value being used to determine the real amount of air injected into the gasifier together with the biomass substance.

Figure 3 demonstrates the effect of ER on CGE after the separators, where increasing ER from 0.15 to 0.45 drives a rise in the efficiency of cold gas after reaching an optimum value, which then reduces constantly due to the decrease in syngas LHV after 0.25 ER, see Equation (2). The ideal case we achieved was at 0.25 ER, which was around 42% of CGE.

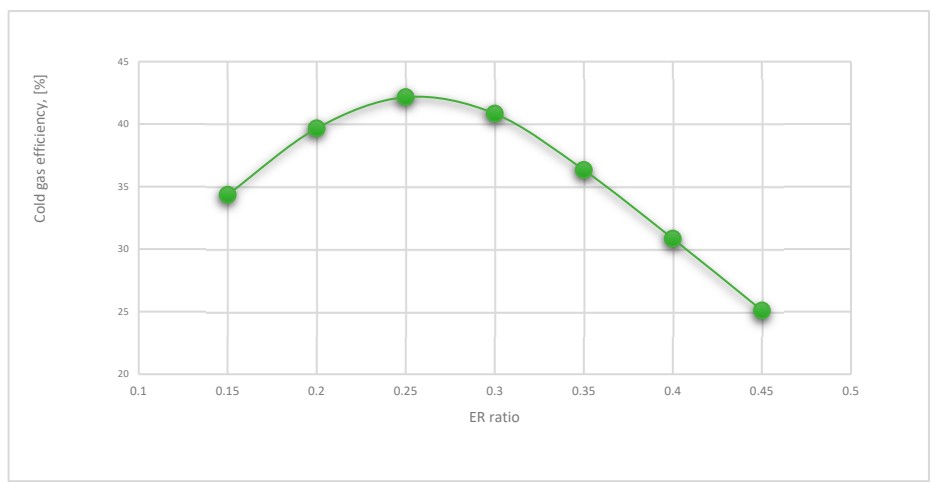

**Figure 3.** Variation of the CGE according to the ER ratio.

As observed in Figure 4 below, the line of injecting real air into the reactor increases continuously depending on the boost of equivalent ratio, where real air basically relies on the ER at which stoichiometric air value was constant, see Equation (1). The case where ER was equal to 0.25 generated a real air amount of almost 37,879 kg/h, which was optimal regarding CGE results.

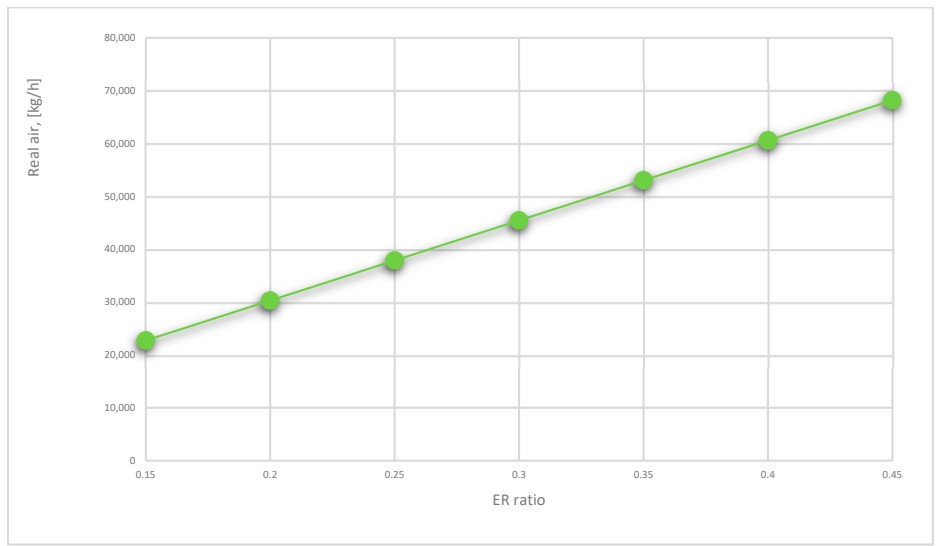

**Figure 4.** Variation of real air introduced into the gasifier regarding several equivalent ratios.

2. The membrane $CO_2$ capture process

Three units of membrane were used for the present paper, where the main varied parameters utilized were $MSA_1$ and $CP_{1,2,3}$. The results demonstrated a high effect of increasing the first membrane surface on raising the capture rate and power consumption. The first compressor has the biggest hand in influencing $CO_2$ capture efficiency, the energy required for the whole process, and carbon dioxide capture purity. The second compressor demonstrated a considerable impact on the efficiency of the second membrane, as described minutely in the coming figures. The syngas stream back from the second membrane helps to reduce the amount of $CO_2$ that leaves the process, this increases the capture efficiency. The purity of carbon dioxide was mainly influenced by the third compressor variation due to the particle transfer at high pressure.

The optimum case achieved to produce more than 95% of $CO_2$ purity and carbon capture efficiency of 90% with minimum electrical energy requirements is at 800,000 m$^2$ $MSA_1$ and (4, 2, 2) of $CP_1$, $CP_2$, and $CP_3$, respectively. The power consumed in this case is nearly 19.7 MW. The other parameters (such as syngas content, temperature, etc.) gained in this case were displayed extensively in Figure 2.

All the coming figures were exhibited to present how the membrane process varied regarding the several variations simulated.

Figure 5 below presents the influence of $CP_1$ on carbon capture efficiency regarding different second compressor pressure values (2–6 bar). As can be clearly shown, increasing the first compressor pressure (2–6 bar) helps to raise the amount of $CO_2$ via the first membrane module, which enormously increases $CO_2$ capture efficiency. At 4 and 6 bar of $CP_1$, $CO_2$ capture efficiency is almost 100% after 4 bar of $CP_2$ values because all carbon dioxide content passed through the membrane. Due to the position of the second compressor in the process configuration, the second compressor pressure variation has a poor influence on the overall $CO_2$ recovery rate.

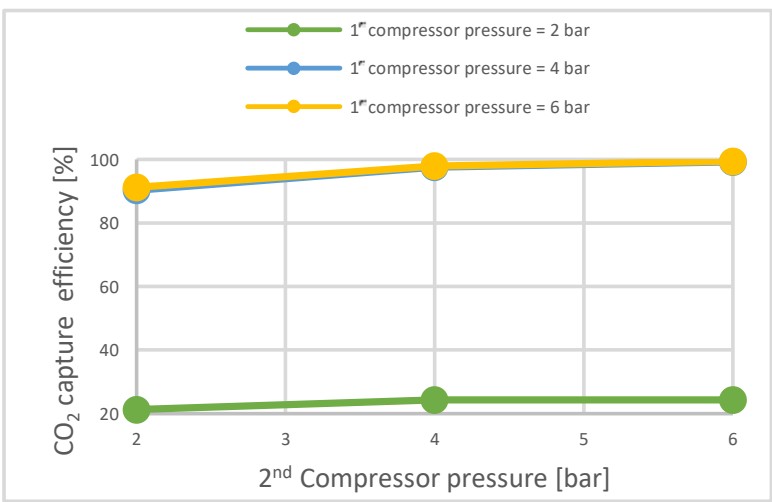

**Figure 5.** First compressor effect on $CO_2$ capture efficiency at different $CP_2$ and 800,000 m$^2$ of $MSA_1$.

As demonstrated in Figure 6, the power consumed in the first compressor has an essential influence on the total energy demanded by the membrane process. That can be explained by the gas stream retentates from the second membrane and the integration with the primary syngas flue, which leads to a rise in the energy needed to push it through the first membrane unit. At 2 bar of $CP_1$, it is visible that increasing the second compressor pressure showed more power requirements due to the flow passing through the second membrane module, which increases the demands to compress it at the third compressor. Due to the low syngas flow recirculated from the first membrane at higher $CP_2$, the flue

stream entering $CP_1$ was reduced, which decreased the power consumption needed at 6 bar of $CP_1$.

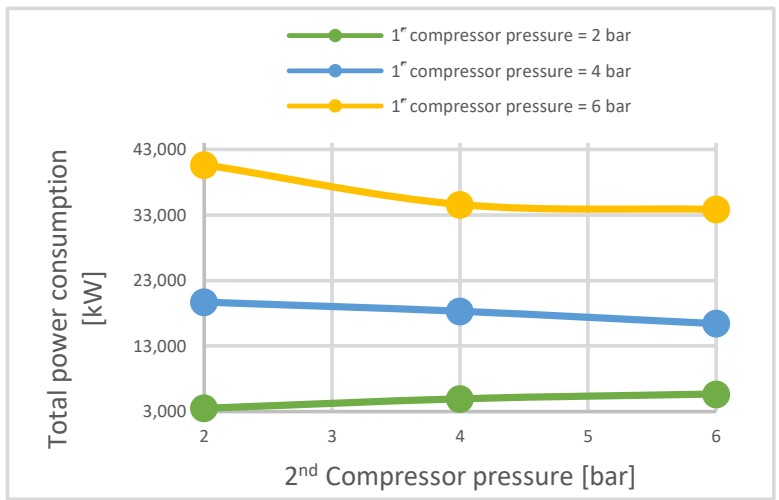

**Figure 6.** Electrical power needed at different $CP_2$ and 800,000 m$^2$ of $MSA_1$.

Figure 7 demonstrates the significance of using many stages of membrane to increase $CO_2$ captured purity, where three stages of membrane produce more $CO_2$ purity than two stages due to the lower membrane surface area used at the third membrane unit (23,000 m$^2$). In terms of $CO_2$ purity, one stage of the membrane is unfavorable for high purity because of the large area of membrane used to achieve high efficiency. As revealed in the figure, the second compressor has a large influence on the second and third membrane units, where increasing the pressure value drives other gases besides carbon dioxide to pass through the membrane.

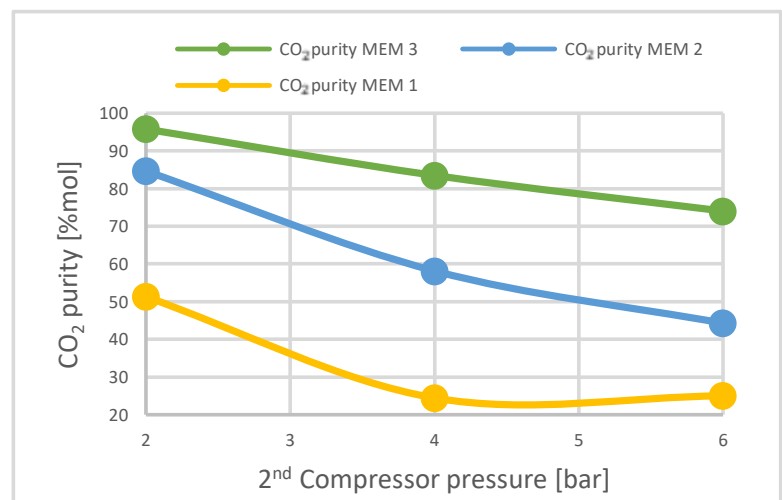

**Figure 7.** Second compressor impact on $CO_2$ purity at 800,000 m$^2$ of $MSA_1$ and 4 bar $CP_1$.

Figure 8 exhibits the extreme action of first membrane surface and compressor pressure on the capture rate. Enlarging the membrane area drives a high boost in syngas flow passing through it, increasing $CO_2$ recovery efficiency. At 1,200,000 m$^2$ and 4 bar of $CP_1$, all the stream passed via the membrane stage, reaching 100% of the carbon rate. At 800,000 m$^2$ $MSA_1$, increasing first compressor values from 2–6 bar resulted in a 78% increase in $CO_2$ recovery rate. The optimal efficiency selected in this paper regarding $CO_2$ purity and power demands is 90% at 800,000 m$^2$ and 4 bar of $CP_1$.

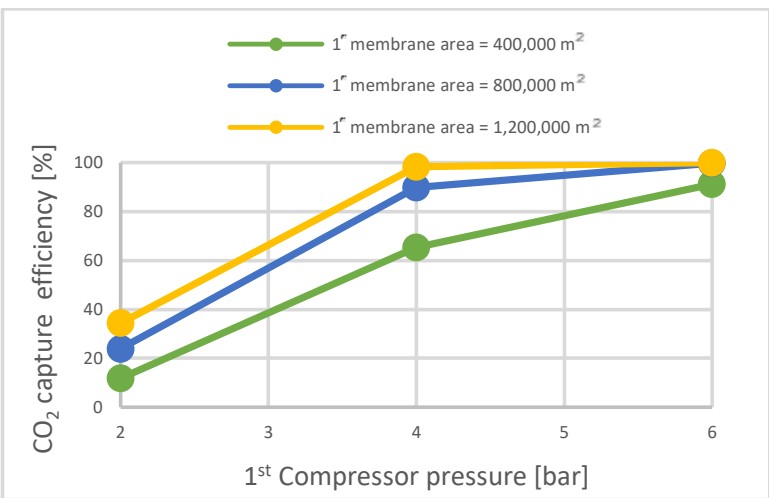

**Figure 8.** $CO_2$ capture efficiency regarding several first membrane surfaces and first compressor pressure.

Regarding Figure 9 below, a larger membrane surface leads to a considerable increase in electricity demand due to the huge syngas flow passing through bigger surfaces, increasing the energy consumption required. The energy demands increased constantly with the raise of the first compressor pressure, because this compressor location was fundamental in that the maximum syngas stream flow passes through it and must be compressed. The energy required for the best status was 19.7 MW at $CP_1$ of 4 bar and 800,000 $m^2$ of $MSA_1$. The reason for not choosing $MSA_1$ of 1,200,000 $m^2$ as an optimum was the elevated demand for electricity compared to using 800,000 $m^2$.

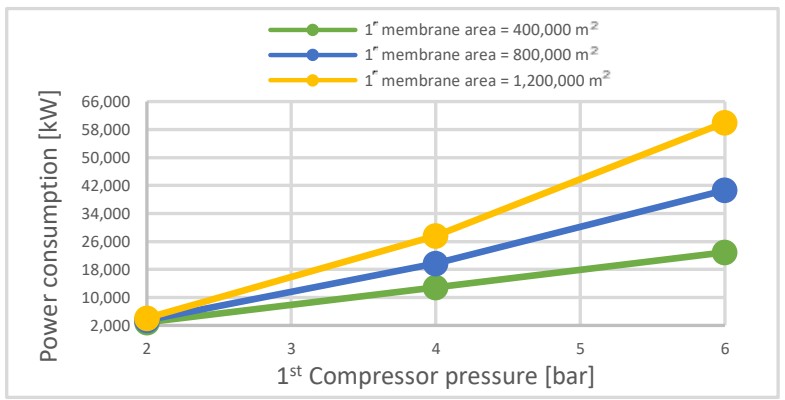

**Figure 9.** Total electrical energy consumption regarding several first membrane surfaces and first compressor pressure.

As presented in Figure 10, all carbon dioxide purity lines went down regarding the boost of the third compressor pressure at different $MSA_1$, where $CP_3$ is the main indicator affecting $CO_2$ purity due to the high stream generated at higher pressures that allow other molecules besides carbon dioxide to pass through the third membrane. The first membrane surface area has a lower influence on $CO_2$ captured purity because of its position far from the third membrane, see Figure 2. The optimum purity chosen for this paper regarding carbon rate and power consumption was more than 95% at 800,000 $m^2$ $MSA_1$ and $CP_3$ of 2 bar.

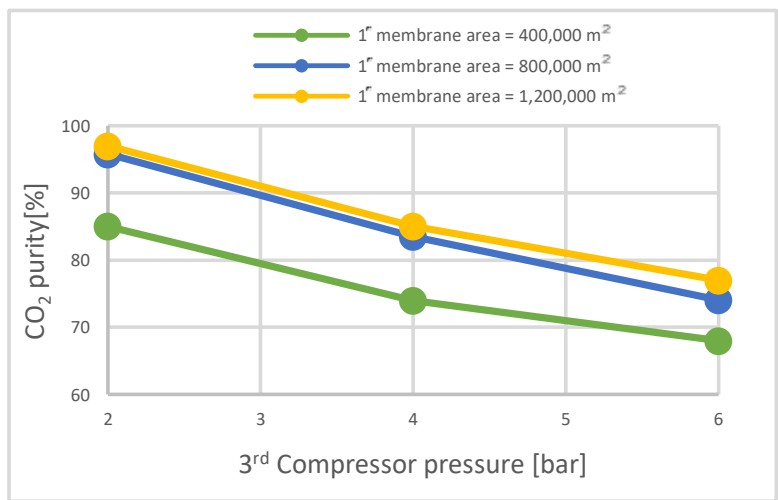

**Figure 10.** $CO_2$ purity variation regarding the third compressor, first membrane surface, and 4 bar of $CP_1$.

Based on $CO_2$ recovery efficiency, purity, and electric energy needed for the process, Table 5 exhibits the ideal values of different membrane surfaces and first compressor pressures at second and third compressor pressures of 2 bar, where the carbon dioxide content that entered the membrane process is 31,524 kg/h.

**Table 5.** The particular indicator values resulting from the simulation.

| First Membrane Area | $m^2$ | 400,000 | | | 800,000 | | | 1,200,000 | | |
|---|---|---|---|---|---|---|---|---|---|---|
| First compressor pressure | bar | 2 | 4 | 6 | 2 | 4 | 6 | 2 | 4 | 6 |
| $CO_2$ capture rate | % | 12.1 | 88.7 | 89.7 | 21.3 | 90.3 | 90.15 | 28.4 | 90.3 | 90.6 |
| $CO_2$ purity | % | 68.7 | 95.7 | 95.7 | 79.9 | 95.8 | 95.7 | 84.5 | 95.8 | 95.7 |
| Electrical energy needed | MW | 2.9 | 12.9 | 22.9 | 3.5 | 19.7 | 40.66 | 4.1 | 27.6 | 60 |
| $CO_2$ recovered/ membrane surface | $kg/m^2 \cdot h$ | 0.009 | 0.07 | 0.07 | 0.008 | 0.035 | 0.035 | 0.007 | 0.022 | 0.023 |

As discussed before, the optimum case selected to capture $CO_2$ flow exits in the syngas stream generated from the plum pits' gasification is at 800,000 $m^2$ of the first membrane surface area and a compressor pressure of 4 bar. This status is technically efficient; therefore, this case was economically analyzed to estimate the specific economic parameters for $CO_2$ capture.

Regarding the perfect case selected for membrane technology (800,000 $m^2$ of $MSA_1$), Table 6 presents an evaluation differentiation between the BIGCC power plant without the utilization of the membrane process.

The moment when the membrane process was combined with the BIGCC power plant, the net energy generated was reduced by about 60% due to the extra power demanded by the auxiliary components used in the membrane (such as compressors). As already mentioned, biomass is a neutral fuel that absorbs $CO_2$ during its growth for photosynthesis, which elucidates why the carbon dioxide recovery factor is minus after utilizing the capture technology. The Integrated membrane process caused a significant increase in LCOE of 69%, which can be explained by the several items used to remove carbon dioxide from the syngas flow.

Table 7 below shows the main economic prediction factors of BIGCC with the optimal case chosen to remove 90% of carbon dioxide with 95% purity (at 800,000 $m^2$ of $MSA_1$), with the membrane capture process relying on Equations (8)–(11).

**Table 6.** The technical and economical estimation of BIGCC with and without membrane process.

| Parameter | Unit | BIGCC Single | BIGCC with Membrane |
|---|---|---|---|
| Introduced biomass | t/h | 31.86 | 31.86 |
| Global efficiency (LHV syngas) | % | 62.20 | 37.60 |
| Global efficiency (LHV biomass) | % | 29.80 | 18.04 |
| Net power produced | kW | 50,000 | 30,245 |
| $CO_2$ recovery factor | kg/MWh | 0.00 | −822.63 |
| $CO_2$ recovered | kg/MWh | n.a. | 939.11 |
| Electricity needed for membrane process | $kW_e$ | n.a. | 19,700 |
| Membrane power consumption | $kWh/tCO_2$ | n.a. | 694 |
| LCOE_rate | EUR/kWh | 0.0974 | 0.1410 |
| SPECCA | $MJ_{th}/kg$ | n.a. | 4.60 |
| SEPCCA | $MJ_{el}/kg$ | n.a. | 2.86 |
| $CO_2$ avoided price | EUR/t | n.a. | 52.94 |
| $CO_2$ captured price | EUR/t | n.a. | 46.37 |

**Table 7.** The cost estimation for BIGCC with the optimum case of membrane process integration.

| Indicator | Unit | Value |
|---|---|---|
| NPV | MEUR | 98.32 |
| IRR | % | 11.6 |
| DPP | year | 14.7 |
| PI | - | 1.32 |

The fine cost of the whole power plant with membrane utilization is 98.32 MEUR, where the annual rate of expansion is almost 12%. Regarding the table, the project is expected to recover its investment cost in 14.7 years. The profitability index demonstrated that the present project is profitable where its value was more than one (1.32).

Figure 11 presents the cumulative cash flow of the project during its lifetime, where after almost 14.5 years, the investment cost will be recovered, and the following years can be considered as achieving a profit.

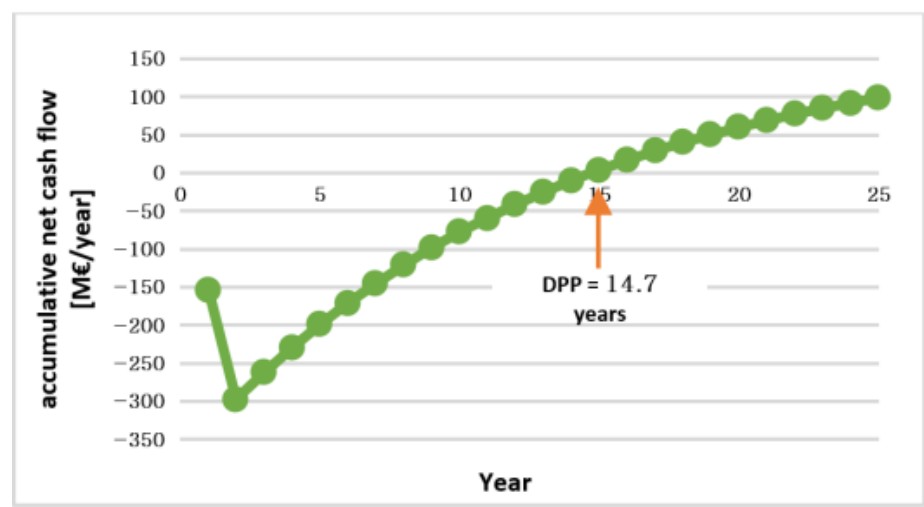

**Figure 11.** The DPP of the accumulative net cash flow.

Figure 12 below presents the impact of the costs of CAPEX, fuel, and other different parameters on the levelized cost of electricity of BIGCC with the optimal case of membrane technology. The effect of the CAPEX and plant capacity factor on the LCOE is salient, where LCOE varies from almost 125 to 155 EUR /MWh by changing CAPEX cost ± 10%.

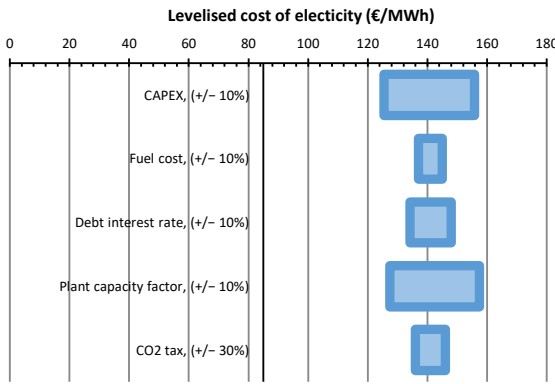

**Figure 12.** LCOE variation regarding several factors.

To present a net perception concerning a model that utilizes the CHEMCAD program with membrane technology, Table 8 below shows a detailed comparison between our optimum case and other articles already published regarding membrane performance respecting various substantial parameters.

**Table 8.** The comparison of the recent optimum case and different articles concerning technical and economical parameters.

| Parameters | Optimum Results for the Current Study | Research from Literatures | | |
|---|---|---|---|---|
| | | [22] | [60] | [61] |
| Number of stages | 3 | 2 | 2 | 2 |
| $CO_2$ capture efficiency, [%] | 90.3 | 90.0 | 79.0 | 84.2 |
| $CO_2$ purity, [%] | 95.8 | 95.0 | 68.0 | 93.6 |
| Total membrane surface, [$m^2$] | $9 \times 10^5$ | n.a. | $6.1 \times 10^5$ | $71 \times 10^5$ |
| $CO_2$ permeance, [GPU] | 1000 | 2000 | 100 | 270 |
| $CO_2/N_2$ selectivity | 50 | 70 | 43 | 41 |
| Flue gas, [kmol/h] | n.a. | 118,694.3 | 52,929 | 65,486 |
| Syngas flow, [kmol/h] | 3109 | n.a. | n.a. | n.a. |
| $CO_2$ content in the stream before membrane, [kmol/h] | 716.29 | 16,296.73 | 6880.77 | 9823 |
| Power consumption of membrane plant, [kWe] | 19,700 | 261,100 | n.a. | n.a. |
| LCOE_tax, [EUR /kWh] | 0.1410 | n.a. | n.a. | n.a. |
| SPECCA, [$MJ_{th}$/kg] | 4.60 | n.a. | n.a. | n.a. |
| SEPCCA, [$MJ_{el}$/kg] | 2.86 | 1.66 (calculated) | n.a. | n.a. |
| $CO_2$ avoided cost [EUR/t] | 52.94 | n.a. | n.a. | 46.0 |
| $CO_2$ captured cost [EUR/t] | 46.37 | 45.10 | 48.01 | n.a. |

Validation of the results was performed by comparing the results obtained for the optimal case (the current research) with the results obtained in the literature (Table 8). It is observed that the membrane performance of the present work shows different technical and economical results compared to the other works in the literature, due to different gas fluxes, different $CO_2$ content, but also due to different number of steps of the membrane system. In the present paper, by using a number of three steps, the SPECCA indicator value of 2.86 $MJ_{el}$/kg was obtained. Based on the data provided in the paper [60], the SPECCA indicator value is 1.66 MJ/kg lower than in our case due to higher permeability (2000 versus 1000 GPU) and higher $CO_2/N_2$ selectivity. These parameters strongly influence the energy consumption required by the membrane. The authors found in [60] that the $CO_2$ captured cost is slightly less than ours, which can be explained by a higher $CO_2$ permeance (2000 GPU) which reduced the electric energy requirements. In the reference paper [61], the $CO_2$ avoided cost is lower than the current optimum result due to the low carbon dioxide recovery achieved (84.2%).

## 5. Conclusions

This article focused on integrating three units of membrane carbon recovery technology into a super-critical power generation station of 50 MW, utilizing plum pits feedstock biomass as a main fuel. Many of the equivalent ratios were simulated in the gasification process to set the optimal cold gas efficiency that defined the flow rate entering the membrane capture procedure. Several parameters were examined regarding the membrane method to capture 90% of the total $CO_2$ emissions and purity of more than 95% with the lowest possible electricity needed for the process.

The equivalent value (ER) can be considered a major factor affecting the cold gas efficiency, where increasing the value from 0.15 to 0.25 showed a bigger CGE of 19% (optimum). When we raised that rate to 0.45, CGE was reduced to 60% because of the mitigation of LHV at an ER of more than 25%. On the other hand, the actual air amount entering the gasification process depends on ER. The results demonstrated a boost of almost 32% of actual airflow if ER raises by 0.15–0.45.

Furthermore, the first compressor pressure is the master component that manipulates $CO_2$ recovery rate, the energy required, and the purity of the $CO_2$ removed. The score values presented that excess $CP_1$ from 2–6 bar commands the carbon efficiency to a growth of approximately 78%, while it drives a rise in the electricity required of 88% at 800,000 $m^2$ $MSA_1$. $CO_2$ purity is also impacted by the increase in first compressor pressure by around 17% at the same first membrane surface. The first membrane surface has a senior direct influence on the whole recovery rate, where the results exhibited a 37% efficiency high when $MSA_1$ increased 400,000–1,200,000 $m^2$, and this value was gained at 4 bar of $CP_1$. The second compressor directly affects the second membrane efficiency, which leads to a decrease in total power demands of 16% regarding its increase from 2 to 4 bar. In terms of increasing $CP_3$ 2–6 bar, there is an immediate reduction in the purity of the carbon removed of around 36%. Therefore, the main influencer on $CO_2$ purity can be counted as the third compressor unit. The electrical power demand to attain the favorable value of recovery rate (90%) and $CO_2$ purity (95%) was 19.7 MW, and that accounted for about 39% of the overall plant capacity (50 MW).

Integrating three stages of membrane directly impacted the efficiency of BIGCC regarding LHV of the syngas, where it was reduced by approximately 39% for the same amount of fuel feedstock due to energy used for the membrane. The LCOE taxes increased with membrane utilization and can be reduced primarily by reducing the power required for $CO_2$ removal efficiency.

As a future action, $CO_2$ permeability has to be increased (e.g., 3000 GPU) to achieve the same $CO_2$ capture efficiency and purity with less power consumption and membrane surface due to high $CO_2$ content passing through a membrane of higher permeance and selectivity. Increasing $CO_2$ permeability helps to reduce the losses of the global power plant efficiency, and on the other hand, decreasing $CO_2$ avoided, $CO_2$ captured, and LCOE prices as well.

**Author Contributions:** Conceptualization, M.A. and C.D.; methodology, M.A. and C.D.; software, M.A. and C.D.; validation, M.A. and C.D.; formal analysis, M.A. and C.D.; investigation, M.A. and C.D.; resources, M.A. and C.D.; writing—original draft preparation, M.A. and C.D.; writing—review and editing, M.A. and C.D.; visualization, M.A. and C.D.; supervision, C.D.; project administration, C.D.; funding acquisition, C.D. All authors have read and agreed to the published version of the manuscript.

**Funding:** The study was funded by the UEFISCDI within the National Project 106PTE/2022—CAPSOFT. Additionally, the research leading to these results received funding from the NO Grants 2014–2021, under project contract No. 13/2020.

**Institutional Review Board Statement:** Not applicable.

**Informed Consent Statement:** Not applicable.

**Data Availability Statement:** Not applicable.

**Conflicts of Interest:** The authors declare no conflict of interest.

## Nomenclature

| | |
|---|---|
| BIGCC | Biomass Integrated Gasification Combined Cycle |
| HRSG | Heat Recovery Steam Generator |
| WGS | Water Gas Shift Reactor |
| CCS | Carbon Capture and Storage |
| ER | Equivalent Ratio |
| CGE | Cold Gas Efficiency |
| LHV | Lowest heating value |
| $CP_1$ | 1st Compressor Pressure |
| $CP_2$ | 2nd Compressor Pressure |
| $CP_3$ | 3rd Compressor Pressure |
| $P_{ax}$ | Energy required for auxiliaries |
| $MSA_1$ | First membrane Surface Area |
| LCOE | Levelized Cost of Electricity |
| $W_{net}$ | Net electric energy generation |
| $E_{No\ capture}$ | $CO_2$ emissions without CCS |
| $E_{with\ capture}$ | $CO_2$ emissions with CCS |
| SPECCA | Specific primary energy consumption for $CO_2$ avoided |
| $CO_{2,CC}$ | $CO_2$ Capture Cost |
| $CO_{2,AC}$ | $CO_2$ Avoided Cost |
| NPV | Net Present Value |
| $IN_i$ | Actual bonus of the year i |
| $C_i$ | Amount of money required for maintenance for a year |
| $A_i$ | Value of a payback for a year |
| $I_i$ | Actual investment for a year |
| r | Rate of deduction |
| IRR | Internal Rate of Return |
| DPP | Discount Payback Period |
| PI | Profitability Index |

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
