# Peer review of "Parametrization Study for Optimal Pre-Combustion Integration of Membrane Processes in BIGCC"

_sustainability, doi:10.3390/su142416604_

Round 1
Reviewer 1 Report
This paper provides some interesting information about the simulated process for capturing CO2. It’s meaning and essential for our current environment. However, there are still some revisions needed to be made. Here are the comments.
1. Some number, such as 18 939.1 (line 111), 17 139 kJ/kg (Table 1) and 3 109 (ER=25%), seem to miss a comma. Please check the whole manuscript.
2. The information in Table 1 is a little vague. The upper part of the table seems to show the composition of some material. However, there is not any description about this. In addition, the connection between the upper and lower parts was not clear. Please check and revise.
3. Fig. 4 show the three stages of the membrane scheme and some description is also provided. Nevertheless, how the CO2 was captured is not described. Please show the major mechanism or equation of the capturing CO2 to help reader to grasp the points easily.
4. In line 290, “At 4, 6 bar of CP1” is suggested to be “At 4 and 6 bar of CP1“.
5. In “The best efficiency selected in this paper” (line 323), “optimal” might be more suitable than “best”.
6. Table 7 provides some important indicators about the cost estimation for the process integration. However, the description about these indicators is deficient. Please explain them in more detail.
Author Response
Dear Reviewers,
First, we would like to thank you for your worthy comments, which will significantly improve the article.
The manuscript structure has been revised and improved starting from the introduction to the discussion section.
English language improvements have been applied to the paper (Grammar and punctuation).
Methodology section has been checked and enhanced by adding more information about the membrane CO2 capture major mechanism. Page 3
Table 1 on page 3 has been modified, where the lower part of that table has been moved and integrated with table 3 on page 5.
The results and discussion section has been enhanced by adding a small description regarding the processes studied starting from line 260.
The references section has been revised and modified depending on the edited and additional information that was added to the paper.
Note/ The inquiries of the reviewer have been marked with a yellow highlight and the answers with a green one.
Reviewer 1
Q1. Some number, such as 18 939.1 (line 111), 17 139 kJ/kg (Table 1) and 3 109 (ER=25%), seem to miss a comma. Please check the whole manuscript
Answer- All the values have been checked and edited
Q2. The information in Table 1 is a little vague. The upper part of the table seems to show the composition of some material. However, there is not any description about this. In addition, the connection between the upper and lower parts was not clear. Please check and revise.
Answer- Table 1 has been improved, where the upper part presents the biomass waste components as described in the new title of the table and in line 111. Regarding the lower part, it has been added to table (3), after membrane parameters, and titled ‘power plant main parameters’.
Q3. Fig. 4 show the three stages of the membrane scheme and some description is also provided. Nevertheless, how the CO2 was captured is not described. Please show the major mechanism or equation of the capturing CO2 to help reader to grasp the points easily.
Answer- The description of main membrane mechanism to capture CO2 has been added to the paper starting from line 136 to 147.
Q4. In line 290, “At 4, 6 bar of CP1” is suggested to be “At 4 and 6 bar of CP1“.
Answer- It has been revised and enhanced
Q5. In “The best efficiency selected in this paper” (line 323), “optimal” might be more suitable than “best”.
Answer- The word has been revised and changed to “optimal”
Q6. Table 7 provides some important indicators about the cost estimation for the process integration. However, the description about these indicators is deficient. Please explain them in more detail.
Answer- The description of table 7 (on page 12) has been revised and improved with more specific detail.
Reviewer 2
Q1. The entire paper is talking about the gasification and carbon capture, but it is hard to tell this from the title. Please be more specific about the work you are focusing on within the title, instead of using some general words like “energy technologies”.
Answer- The title of the paper was revised and upgraded to “Parametrization study for optimal pre-combustion integration of membrane processes in BIGCC”
Q2. The structure of this manuscript needs to be improved. Please focusing on the topic, try to get rid of some contents which is not necessary. For example, as I understand, Figure 2 is enough to clearly show the structure of the energy system. Readers can easily get lost when seeing figure 1 – figure 3. If I am right, Figure 1 and figure 3 are not the systems which will be focused in this work
Answer- The structure of the manuscript has been revised and modified. Figure 1 and figure 3 have been removed and figure 2 was modified and became figure 1 on page (3)
Q3. Please unify the format of equations, they look like a little bit messy
Answer- All the equations format has been revised and unified. Equation 5 can’t be unified with others because it is long.
Q4. The quality if all the figures must be improved, it is hard for reading.
Answer- We followed the template of the sustainability journal. However, if they will ask to improve the equality of figures we are able to do that. Anyway in this comments file please find below the figures improved.
Figure 1. Scheme diagram of BIGCC with pre-combustion carbon capture.
Figure 2. Three stages of the membrane scheme with different components.
Q5. Authors preformed a series of studies to investigate the impacts of different parameters on the system performance. But those studies are not well designed. Normally, a comprehensive sensitivity analysis and some optimization works are required for the performance improvement of this kind of energy systems.
Answer- The subject focuses on developing a technical and economical assessment of membrane technology taking into account the experimental results obtained by SINTEF in the CO2 hybrid project where the UPB is the coordinator. Anyway, we include figure (12) at page 13 in the paper to present the sensitivity assessment considering the LCOE variation based on the range where the main parameters were varied.

Reviewer 2 Report
The authors presented a research on the techno-economic analysis of a gasification process with pre-combustion carbon capture with membrane systems. The topic of the manuscript is within the scope of sustainability, but at least a major revision is required before it can be published in the journal. Here are some detailed comments:
1. The entire paper is talking about the gasification and carbon capture, but it is hard to tell this from the title. Please be more specific about the work you are focusing on within the title, instead of using some general words like “energy technologies”.
2. The structure of this manuscript needs to be improved. Please focusing on the topic, try to get rid of some contents which is not necessary. For example, as I understand, Figure 2 is enough to clearly show the structure of the energy system. Readers can easily get lost when seeing figure 1 – figure 3. If I am right, Figure 1 and figure 3 are not the systems which will be focused in this work.
3. Please unify the format of equations, they look like a little bit messy
4. The quality if all the figures must be improved, it is hard for reading.
5. Authors preformed a series of studies to investigate the impacts of different parameters on the system performance. But those studies are not well designed. Normally, a comprehensive sensitivity analysis and some optimization works are required for the performance improvement of this kind of energy systems.
Author Response

(The authors gave the same response as above.)

Round 2
Reviewer 2 Report
Authors have made a significant revision on their manuscript. But, the major issues have not been addressed yet. the research performed in this work in not organized in a scientific way.
For the performance improvement or performance investigation research of kind of energy systems like this. A sensitive analysis of all the design parameters need to be done first to find the parameters that have big impacts on the system performance. Then, normally a optimization process would followed to get the optimal design for the system. But in this manuscript, it is hard to tell how this kind of procedures are conducted, seems like all the discussion are based on the modeling results based on the random combination of design parameters, and are not organized in a scientific and systematic way. There are also some other concern about the figures, they are not well prepared. Even the font and font size are not unified in a single figure.
Author Response
Dear Editor,
We would like to thank you very much for the comments received. Please find attached the file with our comments.
Sincerely,
Prof. Cristian Dinca

Round 3
Reviewer 2 Report
All the comments have been addressed in the revised version.
Author Response
Dear Editor,
Thank you very much for the wholly revised paper. Accordingly, I have checked all references, and the self-citations have been reduced from 17 to 9 self-citations. All changes are marked in blue on the paper in the references area.
Due to the duplication of Ref [22] with [60], I have removed reference [60].
Sincerely,
Prof. Cristian Dinca
